# Enhancing climate resilience with proximal cues in personalized climate disaster preparedness messaging

**Nurit Nobel** [1,2] ✉ **& Michael Hiscox**[2,3]

Climate-related disasters such as wildfires and floods pose escalating risks to communities worldwide, yet motivating individuals to adopt protective measures remains a persistent challenge. In a pre-registered field experiment with 12,985 Australian homeowners in wildfire-prone areas, we demonstrate that a simple behavioural intervention—integrating proximal cues, such as participants' suburbs, into climate risk communications— significantly increases engagement. Participants who received localized messages were twice as likely to seek further information about wildfire preparedness compared with those who received generic communications (odds ratio of 2.03, 95% confidence interval of 1.33 to 3.16). This effect highlights the power of behavioural interventions in addressing barriers to climate adaptation, particularly by reducing psychological distance and fostering place attachment. By making abstract climate risks tangible and personally relevant, the intervention nudges individuals towards action. These findings suggest a scalable, low-cost approach for enhancing disaster preparedness, offering insights for leveraging behavioural science to mitigate the impact of climate-related disasters.

Climate change is ushering in a new era of heightened environmental risks, characterized by more frequent and intense extreme weather events such as wildfires, droughts, floods and hurricanes[1]. In recent years, the magnitude and occurrence of these events have increased substantially. For instance, while the average number of climate disasters causing damages exceeding US$1 billion annually in the USA was 7.2 between 1980 and 2019, this figure nearly tripled to 20.4 events a year in recent years (2019–2023)[2]. Concurrently, the mean time elapsed since the last wildfire has decreased steadily over the past decades, and the frequency of forest megafire years (defined as years in which more than 1 million hectares burned) has markedly surged[3]. To note, these data do not include the January 2025 Southern California wildfires, which, as of this writing, have burned over 57,000 acres, destroyed more than 18,000 structures and displaced over 200,000 people[4].

This growing prevalence of climate- and weather-related disasters is expected to continue, driven by rising global temperatures, a drying climate and increasing population and urban development in high-risk areas[5,6]. These disasters not only strain local governments and fiscal stability[7] but also impact homeowners in vulnerable areas, where property values may be adversely affected by climate risk[7,8]. While households cannot directly influence the trajectory of climate change, they can take steps to mitigate their vulnerability and minimize the physical and financial damage caused by climate-related stresses. Protective measures include seeking information about climate risks and mitigation strategies, and strengthening homes against potential disasters[9], and strategies promoting adaptive climate resilience have been proposed to mitigate risks for both communities and individuals[6]. However, efforts to encourage voluntary adaptation among homeowners, including complying with building codes and renovating

[1]Center for Sustainability Research, Department of Marketing and Strategy, Stockholm School of Economics, Stockholm, Sweden. [2]The Sustainability Transparency Accountability Research Lab, Weatherhead Center for International Affairs, Harvard University, Cambridge, MA, USA. [3]Department of Government, Harvard University, Cambridge, MA, USA. ✉e-mail: nurit.nobel@hhs.se

properties, have largely been ineffective, highlighting the urgent need for innovative approaches[10].

Mobilizing individuals to adopt climate-conscious actions and behaviours is a topic that has increasingly occupied stakeholders and researchers across disciplines[11–13]. Researchers in behavioural science have advocated for taking human decision-making and psychological factors into account when designing interventions to encourage pro-environmental behaviour[11,14–17]. Behavioural interventions have been tested to encourage a myriad of behaviours aimed at reducing carbon emissions[18,19], such as choosing greener product alternatives[20,21], switching to sustainable diets[22,23], recycling[24,25] and shifting commuters away from single-occupancy vehicles[26,27].

While household decarbonization is essential to curb the harmful effects of climate change[28], there has been a paucity of research exploring how behavioural science can tackle an equally important climate challenge: encouraging consumers and homeowners to engage in behaviours that increase their properties' climate resilience. We argue that getting consumers to maintain and develop their properties to adapt to looming climate risk represents a critical climate challenge that has yet to be tackled by behavioural interventions, despite the potential benefits in applying this approach. The current research contributes to this under-explored area of research, by testing a behaviourally informed intervention to encourage consumers to prepare and adapt their property in order to mitigate risk of wildfires ahead of the summer season in Australia. The intervention aimed at increasing the tangibility of the message[16], using a proximal cue to reduce psychological distance[29] and invoke place attachment[30]. We tested the intervention in a large-scale pre-registered field experiment with 12,985 homeowners whose properties are located in wildfire prone areas.

## Behavioural barriers to climate resilience

Promoting climate resilience faces similar challenges to promoting decarbonization behaviours. Chief among these is the extended time required for outcomes to be realized. While decarbonization behaviours promise distant future payoffs associated with reduced temperatures and fewer extreme weather events among other benefits, climate resilience behaviours focus on risk mitigation, such as avoiding or reducing potential future losses from wildfires or floods on one's property. Despite their different potential benefits, both decarbonization and resilience behaviours share the key characteristic that their positive consequences are not incurred immediately but instead manifest years into the future. Unfortunately, this has an adverse effect on consumers' propensity to engage in these behaviours, since copious research has shown that future payoffs are less attractive to consumers than immediate ones[31–35].

The long time horizon challenge for promoting sustainable behaviours is exacerbated by the nature of the consequence themselves, which are often perceived by consumers to be abstract, vague and uncertain[36,37], while the costs involved in engaging in the pro-environmental behaviour are both immediate and concrete. In the context of climate resilience, protecting a home from wildfires involves, among other tasks, the clearing of gutters and increased lawn and yard maintenance to minimize susceptibility to fire. While these actions require time and effort right now, their benefits are both uncertain and incurred far into the future, if at all. The main payoff in this case would be safeguarding the consumer from the potential loss in home value owing to wildfire, though the extent of this loss, the likelihood of a fire and the effectiveness of the preventive measures are all uncertain.

The long-term, fuzzy, often far-removed nature of climate issues presents a barrier for action, which is rooted in the concept of psychological distance. Based on construal level theory, psychological distance refers to the extent to which objects are removed from a person's immediate self hypothetically (real or fiction), temporally, spatially or socially[38,39]. The more psychologically distant an object, person, place or event, is perceived to be, the more abstract and high-level its mental representation becomes. The construal of these psychologically distant objects stays at the 'big picture' level rather than focusing on details. Psychological distance can therefore affect a person's perception of an object or event, as well as attitudes and responses towards it[30].

This idea of psychological distance can be applied to attitudes and behaviours concerning climate change and climate resilience. The extent to which people would be willing to change and adopt pro-environmental behaviours depends on whether they perceive climate change to be a concrete phenomenon that closely affects their own lives, that is, psychologically close[29]. The effects of psychological distance on people's perceptions of climate change can manifest in various ways[40]: individuals may doubt that climate change is occurring at all, or if they acknowledge its existence, they may question the extent of its threat[41], indicating a hypothetical psychological distance. They may accept that climate change is happening and with severe impacts, but still feel psychological distance from it because they expect the implications to unfold far into the future[42], reflecting temporal psychological distance. Alternatively, they may accept that climate change is happening and is already affecting people, but still question whether its effects would be felt in their own immediate sphere[43], illustrating spatial psychological distance, and whether it affects people like them[44], representing social psychological distance.

Spatial psychological distance seems particularly relevant to the challenge of encouraging climate action, as it also relates to a person's place attachment, the affective and cognitive ties formed between an individual and a specific place[30]. A strong connectedness to a place can breed protective feelings towards this place, and thus affect climate attitudes and behaviours[45]. When individuals care strongly about a place, they are more likely to want to protect it, and more willing to exert effort to do so[46,47]. Furthermore, when individuals recognize that a global crisis such as climate change has local consequences, they may become more attuned to the risks involved which increase in salience[30].

That said, the idea that psychological distance is a central barrier to climate action has been called into question. One review has claimed that psychological distance-based interventions have not been universally effective for encouraging climate mitigation, and that their efficacy is conditional[48]. A more recent review argues that the psychological distance of climate change may be overestimated and may not predict climate-related beliefs and behaviours as strongly as previously assumed[49,50]. However, recent empirical findings from a large-scale mega-study with 59,440 participants from 63 countries, provide evidence supporting the continued relevance of interventions aimed at reducing distance[12,51]. In the mega-study, an intervention to decrease proximal psychological distance emerged as the most effective out of 11 expert-sourced interventions for increasing belief that climate change is a threat to society. Moreover, the intervention that surfaced as the most effective for increasing policy support aimed at reducing another type of psychological distance—social psychological distance—by fostering inter-generational self-continuity[12]. These findings underscore that climate change can still be perceived as distant across multiple dimensions (temporal, spatial, social and hypothetical), and that interventions that aim to make it feel psychologically closer remain promising and empirically supported tools for promoting climate action.

## Using proximal cues in climate communications

In line with the findings on psychological distance, research has shown that personal experience of climate change, whether direct or via media reporting, heightens individuals' perception of local climate risks[37,52,53], and that experiencing the perceived effects of climate change firsthand can increase the likelihood of taking climate action[54]. Therefore, to address the challenge of the abstract and long-term nature of

**Table 1 | Summary of sample characteristics**

| | Total sample (*n*=12,985) | Control (*n*=6,524) | Treatment (*n*=6,461) | *P* value |
|---|---|---|---|---|
| **Demographics** | | | | |
| Gender | 49.4% | 49.2% | 49.6% | 0.737 |
| Age | 47.1 (12.5) | 47.2 (12.4) | 47.0 (12.5) | 0.330 |
| Annual income | 77,414 (63,526) | 78,938 (71,213) | 75,917 (54,911) | 0.110 |
| **Customer characteristics** | | | | |
| Bank tenure | 21.4 (11.8) | 21.4 (11.8) | 21.3 (11.8) | 0.535 |
| Checking balance | 65,952 (205,080) | 64,317 (207,211) | 67,604 (202,908) | 0.361 |
| Credit card balance | 2,241 (5,331) | 2,249 (5,401) | 2,232 (5,260) | 0.853 |
| Personal loan balance | 619 (4,611) | 639 (4,756) | 600 (4,461) | 0.625 |
| Home loan balance | 593,503 (710,191) | 593,419 (740,394) | 593,589 (678,384) | 0.989 |

Mean values of measures are reported with standard deviations in parentheses. Gender is coded as '1' for female and '0' for not female. Age and bank tenure are measured in years. Annual income and account balances are given in Australian Dollars (AUD).

climate change, increasing the tangibility of climate communications has been proposed in order to boost the concreteness and relevance to the individual[36,55].

One way to increase tangibility is to root climate appeals in a specific place. For example, in a classic study on encouraging towel reuse at hotels, the authors found that the social norms-based appeals were more effective when they mirrored the immediate setting of the individual, referred to as provincial norms ('the majority of guests in this room reuse their towels')[56]. Similarly, a British governmental agency used this technique to increase tax compliance by referring to a localized social norm relevant to the individual targeted ('9 out of 10 people in your town pay their tax on time')[57].

Despite the promise that using proximal cues for increasing tangibility of climate communications has shown, there is limited empirical research applying these interventions to mobilize towards climate adaptation. Existing behavioural science research on psychological distance has overwhelmingly focused on climate mitigation (for example, carbon reduction and sustainable consumption). Field experiments applying this framework to the domain of climate adaptation—particularly to protective homeowner behaviours in the face of acute environmental risks—remain limited. In general, research testing behavioural interventions to encourage the safeguarding of properties to mitigate the effects of a future natural disaster has been scarce, with one notable exception. In that study, researchers showed participants vivid images of what a local school would look like after a major earthquake, to appeal to individuals' emotional, rather than rational, decision-making, which resulted in increased willingness to sign a petition to support seismic upgrades[58]. However, large-scale field experiments testing behavioural interventions for climate-related disaster adaptation remain under-explored in literature. There is evidently a substantial opportunity for further research in this area.

To test whether reducing psychological distance through localized messaging could promote climate preparedness, we conducted a large-scale pre-registered field experiment in partnership with a major Australian bank. In the weeks leading up to the 2023 summer wildfire season, we randomly assigned 12,985 bank customers living in wildfire prone areas to receive either a generic or a suburb-personalized email message about property protection (see Table 1 for summary of sample characteristics). Aiming to reduce psychological distance by evoking place attachment, the intervention introduced a proximal cue to communication on wildfire property safeguarding measures. We then tracked participants' engagement with the message using behavioural measures including clicks and visits to a landing page with further information about wildfire preparedness. Our hypothesis was that including this proximal cue would increase propensity to engage with the climate resilience message.

## Results

To gauge the effectiveness of the messages, we examined participants' engagement with the communication sent from the bank, as a proxy for their willingness to learn about protecting their homes from wildfires. Engagement was operationalized using three behavioural indicators: whether the participant opened the email, clicked a link to learn more or visited the landing page (visits capture both clicks and manual visits, that is, a customer navigating to the landing page via copying the link in the email). In what follows, we report all three measures, while acknowledging important limitations in two of them, namely, opening rates are somewhat diffused, as some email servers in the system used by our field partner record emails automatically as 'opened' regardless of whether they were opened by the user. This anomaly leads to inflated opening rates, although because of randomization it does not affect our main analyses as both control and treatment conditions are equally affected by this. Additionally, the page visits variable is also affected by a peculiarity, whereby not all clicks on the landing page link are registered as page visits, for example when a customer is not previously logged in to the bank portal. Similar to opening rates, this deviation is likewise not expected to affect the main analyses owing to successful randomization. We proceed by reporting the analyses for the three outcome variables, treating landing page clicks as the main outcome variable, as it does not suffer from any of the aforementioned shortcomings. Although we pre-registered visits as a primary outcome owing to its comprehensiveness, we focus on clicks given the data incompleteness issues described above.

All three outcome variables were coded as binary dummy variables where '1' represents a click/visit/open, and '0' represents no click/visit/open. All analyses reported are negative two-tailed binomial regressions, appropriate in the case of skewed distributions. Linear regression results are reported in Appendix D in Supplementary Information and are consistent with the main results presented in the manuscript.

### Landing page clicks

The results show that the presence of the proximal cue in the form of a participant's suburb significantly affected the willingness to learn more about bushfire preparedness, as measured by the propensity to click on the 'learn more' link. While 0.96% of participants in the treatment group clicked on the link to the landing page at the bottom of the email, only 0.48% of participants in the control group did so (see Table 2 for a summary of descriptive statistics for all outcome variables). A negative binomial regression found the effect of condition on the propensity to click on the link to be significant standardized regression coefficient $B = 0.71$, odds ratio (OR) of 2.03, s.e.m. of 0.22, 95% confidence interval (CI) of 0.284 to 1.152, $Z = 3.21$ and $P = 0.001$), meaning

**Table 2 | Summary of outcome variables' descriptive statistics**

| | Control (n=6,524) | | Treatment (n=6,461) | | Total sample (n=12,985) | |
|---|---|---|---|---|---|---|
| | **N** | **%** | **N** | **%** | **N** | **%** |
| Clicks | 31 | 0.475 | 62 | 0.959 | 93 | 0.716 |
| Visits | 4 | 0.061 | 19 | 0.294 | 23 | 0.177 |
| Opens | 3,473 | 53.234 | 3,548 | 54.914 | 7,021 | 54.071 |

The table reports the number (N) and percentage (%) of participants who clicked on the link, visited the landing page or opened the email. All outcome variables were binary ('1' for yes and '0' for no).

that participants in the proximal cue treatment group were around twice as likely to click on the link to the landing page than participants in the control group, and that this effect was significant. The visualization of group differences is shown in Fig. 1.

**Landing page visits**
The results likewise demonstrate that including a proximal cue in the headline significantly affected engagement when measured by visits to the landing page. The rate of page visits in the treatment (proximal cue) condition was 0.294%, whereas it was only 0.061% in the control (no proximal cue) condition (Table 2). A negative binomial regression showed that this difference is significant ($B = 1.57$, OR of 4.81, s.e.m. of 0.551, 95% CI of 1.81 to 16.65, $Z = 2.853$ and $P = 0.004$), meaning that participants in the proximal cue treatment group were around 4.8 times as likely to visit the link to the landing page than participants in the control group, and that this effect was significant. Differences between conditions are visualized in Fig. 2.

**Email openings**
Finally, the results also show a similar trend for including a proximal cue in the communication header when it comes to email opening rates. More people in the treatment group, which included proximal cues, opened the email than people in the control group (54.9% versus 53.2%; Table 2, and visualization of differences between conditions in Fig. 3). The negative binomial regression did not yield statistically significant effects, and therefore does not support the tested hypothesis ($B = 0.07$, OR of 1.07, s.e.m. of 0.35, 95% CI of −0.999 to 1.15, $Z = 1.92$ and $P = 0.055$). To further interpret the null results, we conducted an equivalence test[59]. We pre-specified a smallest effect of interest of ±10% on the odds ratio scale (region of practical equivalence of log(0.90, 1.10) = (−0.105, 0.095)), reflecting measurement noise from auto-opens and the weaker link between opens and downstream outcomes. The condition effect's 90% CI (log-odds of (0.01, 0.13) and OR ≈ (1.01, 1.14)) was not fully contained within the region of practical equivalence ($P = 0.142$ and second generation $P$ value of 0.825). Thus, we cannot conclude practical equivalence under ±10%. Combined with the non-significant null hypothesis significance testing result, the effect is best characterized as inconclusive (neither detectably different from zero nor demonstrably negligible within ±10%).

To summarize, we examined three measures of engagement with a message on bushfire preparedness: landing page clicks, landing page visits and email openings. The results of the main outcome variable of clicks provide support for our hypothesis that communication high in tangibility—such as by highlighting a proximal cue—is more effective in encouraging customer engagement with a climate preparedness message. Our main outcome variable, landing page clicks, showed a statistically significant effect, with visits further supporting this finding. Email openings did not reach statistical significance and should be interpreted accordingly. Additional exploratory analyses for potential moderators did not yield significant results and are reported in Appendix E in Supplementary Information.

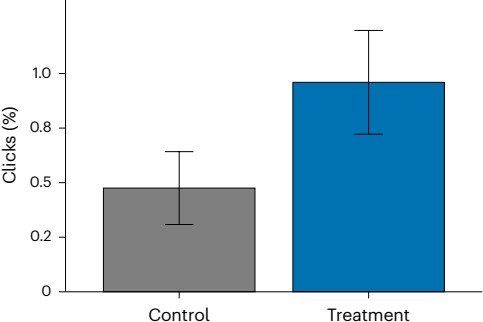

**Fig. 1 | Click-through rate by condition.** The plot shows the mean proportion of participants who clicked the call-to-action in each condition (control $n = 6,524$ and treatment $n = 6,461$). Error bars denote 95% CIs around the mean proportion. Group difference was tested via two-sided negative binomial regression (treatment versus control): $B = 0.71$, s.e.m. of 0.22, OR of 2.03, 95% CI of 1.33 to 3.16, $Z = 3.21$ and $P = 0.001$. No adjustment for multiple comparisons (pre-registered single primary test) was performed. See Table 2 for descriptives.

## Discussion

This large-scale field experiment with Australian homeowners demonstrates the effectiveness of behavioural interventions to encourage climate adaptation measures. Specifically, this study shows that incorporating proximal cues in climate risk communication can enhance engagement with resilience promoting actions. The intervention increased participant engagement with wildfire preparedness messages, as evidenced by significantly higher click-through rates and site visits when communications included localized references to participants' suburbs. These results provide evidence that tailoring messages to reduce psychological distance and foster place attachment can encourage climate resilience.

Although the increase in engagement observed in our study was statistically robust—with the odds of clicking nearly twice as high in the proximal cue group—the absolute change in behaviour was modest (0.959% in the treatment group versus 0.475% in the control group). This pattern is not uncommon in large-scale field experiments, which often yield smaller, yet more ecologically valid, effect sizes compared with controlled laboratory studies[60–63]. However, small effects in behavioural science can still carry substantial practical value[64–67]. Moreover, an exclusive focus on large effect sizes can hinder a nuanced exploration of complex psychological phenomena, and may lead to the dismissal of effects that are both real and meaningful[67–69]. Specifically relevant to our study, the tendency for findings from real-world populations to produce smaller effect sizes than those observed in laboratory studies has been well documented in the context of behavioural interventions[70–72]. But importantly, in real-world settings, even minor behaviour changes can become consequential when implemented at scale—across time, populations and policy channels[70,73]. While our intervention yielded modest effects, it also incurred modest costs, leveraging communication tools the bank is already using. The proximal cue intervention may therefore be considered an effective and scalable tool for behaviour change when evaluated through a cost–benefit lens[66,70,73].

This study makes several key contributions. First, our findings extend the existing literature on psychological distance and climate change by demonstrating that proximal cues can effectively shift individuals' engagement with climate adaptation messages. In doing so, we contribute to the ongoing debate regarding the effects of psychological distance-based interventions, following mixed evidence[48,50]. In the current study, incorporating localized elements into communication increased engagement with climate adaptation messages, thus supporting the view that such interventions can be effective under the right conditions. Importantly, our findings are not at odds with previous

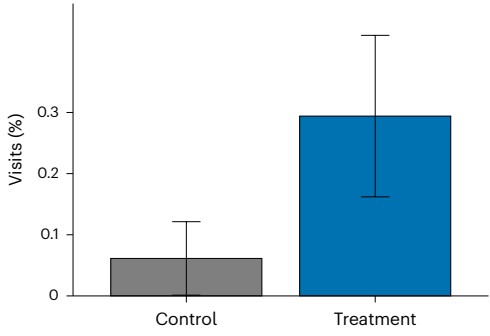

**Fig. 2 | Visit rate by condition.** The plot shows the mean proportion of participants who visited the landing page (control $n = 6,524$ and treatment $n = 6,461$). Error bars denote 95% CIs around the mean proportion. Two-sided negative binomial regression: $B = 1.57$, s.e.m. of 0.551, OR of 4.81, 95% CI of 1.81 to 16.65, $Z = 2.853$ and $P = 0.004$. No multiple comparisons adjustment was performed. See Table 2 for descriptives.

theorizing that challenged the efficacy of proximizing interventions. It has been argued that the success of proximizing interventions is conditional upon the referenced place being meaningful to the recipient, and that the individual feels efficacious in responding to the threat[48]. Our communication was designed to highlight a relevant place for the recipient (the suburb in which they reside), and to increase a sense of self-efficacy by focusing on what the individual can do to mitigate wildfire damage to their home. Thus, the intervention aligns with the enabling conditions outlined by Brügger and colleagues[48]. Moreover, although psychological distance has been widely studied in the context of climate mitigation—such as promoting low-carbon lifestyles and sustainable consumption—its application to climate adaptation remains largely unexplored. Empirical studies testing psychological distance interventions for encouraging homeowners to undertake protective actions in response to acute environmental threats remain limited. Our study contributes to this area by demonstrating that psychological distance-based interventions can be successfully extended to climate resilience behaviour.

Second, this study advances the understanding of the applicability of behavioural interventions in addressing complex societal challenges, particularly in climate adaptation. While behavioural interventions have garnered both enthusiasm and scepticism for their real-world impact[73–77], there is growing recognition that their success hinges on contextual sensitivity[78–82]. Our work contributes by demonstrating that even light-touch behavioural interventions can be impactful at scale by leveraging contextualization and personalization. By timing the message to coincide with the onset of the Australian summer—when wildfire risk is most salient—and personalizing the communication with consumers' suburbs, we enhanced the relevance and immediacy of climate risk. Methodologically, large-scale field studies remain scarce in the domain of climate change mitigation, and few interventions have moved beyond hypothetical settings and self-reported measures[11]. Our intervention contributes by providing evidence from a field setting, studying real homeowners and measuring actual engagement. Taken together, our findings suggest that, when designed with attention to context and incorporating meaningful personalization, behavioural interventions can serve as a powerful tool in climate risk communication and policy outreach. Importantly, they offer a complementary strategy to systemic structural and regulatory measures[83,84], enabling institutions to foster climate resilience behaviour among individuals who might otherwise disengage owing to the abstractness or perceived remoteness of climate threats.

Finally, the results offer actionable insights for practitioners designing effective communication strategies to promote climate resilience and hardening. As homeowners facing climate risks often

encounter rising insurance premiums or even outright denial of coverage[85], implementing climate hardening measures to safeguard their property remains their most effective mitigation strategy. In our study, localized messaging emerged as an effective strategy for amplifying the salience of climate risks and encouraging safeguarding actions. Importantly, the intervention is not only impactful but also cost-effective and scalable, making it an accessible solution for stakeholders such as public agencies, private institutions and community organizations. With the increasing frequency and intensity of climate-related disasters, this approach provides a practical pathway to mitigate potential financial losses. Banks, insurers and local governments, can incorporate proximal cues into their outreach campaigns to ensure messages resonate with individuals and inspire concrete actions that reduce vulnerabilities.

Several limitations of this work should be acknowledged. First, the study measures engagement as a proxy for action, rather than climate resilience actions themselves. With that said, engagement—such as clicking on an informational link—is widely recognized as a necessary precursor to downstream behaviour change, particularly in digital interventions[86,87]. By prompting interest and attention, engagement is a precursor toward intention formation and ultimately behaviour change[88]. Research in the environmental domains has shown that early intention often serves as a first step in the behavioural funnel towards real-world action[89,90]. We believe that measuring clicks as a proxy for engagement provides a higher level of ecological validity than self-reported intentions. However, it would be beneficial if future research could measure completed climate resilience enhancing actions. Second, the study focuses on a specific geographic and cultural context—Australian homeowners in wildfire prone areas—which may limit generalizability. Although we believe these findings would generalize, future research should test the intervention across diverse environmental risks and cultural settings to assess its broader applicability. Lastly, while the use of localized communication reduces psychological distance, it remains unclear whether the same effects would hold in settings where individuals have lower levels of place attachment or awareness of local risks. Future work could investigate the role of these moderating factors in determining the efficacy of proximal cues. Additionally, future research could explore the integration of more specific, localized climate risk information—such as fire risk ratings, recent nearby incidents or neighbourhood-specific forecasts—to further increase message relevance and urgency. Such contextually enriched cues may enhance the salience and perceived credibility of the message, potentially leading to even greater behavioural engagement.

While our results are focused on increasing engagement with wildfire preparedness messages, we believe this research can be

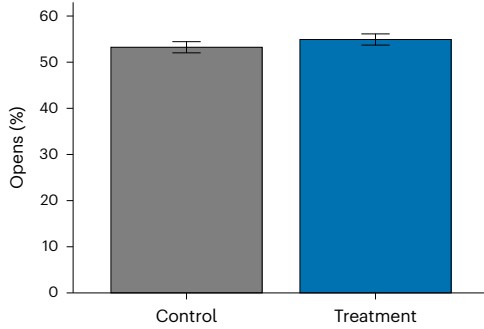

**Fig. 3 | Open rate by condition.** The plot shows the mean proportion of participants who opened the email (control $n = 6,524$ and treatment $n = 6,461$). Error bars denote 95% CIs around the mean proportion. Two-sided negative binomial regression: $B = 0.07$, s.e.m. of 0.35, OR of 1.07, 95% CI of 0.999 to 1.15, $Z = 1.92$ and $P = 0.055$ (not significant). No multiple comparisons adjustment was performed. See Table 2 for descriptives.

extended to addressing a broader range of challenges associated with climate disasters, from flood preparedness to hurricane evacuation planning. Just as targeted behavioural interventions have contributed to public health campaigns[91,92] and promoted financial wellbeing[93–95], integrating localized messaging into climate communication could reshape the way individuals, communities and institutions approach climate resilience. If widely adopted, this approach has the potential to ensure that individuals are not only informed but also empowered to take proactive steps to safeguard their homes from looming climate-related events. In doing so, our research aims to contribute to a future in which communities worldwide are better prepared to face the growing threats of climate change, reducing both human and economic vulnerabilities to natural disasters.

## Methods

To test the effect of including proximal cues in climate risk communication on willingness to safeguard properties, we conducted a field experiment with a total of 12,985 customers of a large Australian retail bank. The experimental design, sample size, exclusions, hypotheses and analyses were pre-registered on 11 December 2023, before data collection. The pre-registration can be accessed at AsPredicted, available at https://aspredicted.org/W75_7DM. The pre-registered plan was followed unless otherwise stated. The Harvard University Committee on the Use of Human Subjects approved the protocol of this study (reference no. IRB23-1716) and determined that informed consent was not required, as is common in the case of field experiments. Participants received no compensation or incentives. The experiment was blinded as participants were unaware that multiple message versions existed, and thus unaware of their own treatment assignment. Randomization, intervention delivery and outcome logging were automated by the partner platform and therefore blind. Data analysis was not performed blind to the conditions of the experiments.

### Setting and participants

Participants were between 19 and 96 years of age (mean $M = 47.1$ and s.d. 12.5), and 49.4% were female. Participants were bank customers of our field partner, who had an active bank account at the time of the intervention (tenure with the bank between 1 and 64 years; $M = 21.4$ and s.d. 11.8) and who were homeowners whose property was located in an area determined to be prone to wildfires (bushfires) in New South Wales (NSW), Australia. Bushfire risk was determined by the Rural Fire Service NSW Bushfire Prone Lands dataset, which is publicly available[96]. In total, the sample size included 12,985 individuals after applying the bank's pre-determined exclusion criteria (for example, minors, customers who had opted out of marketing communications and those participating in other bank campaigns) as pre-registered.

No statistical methods were used to pre-determine sample size, but our sample size is larger than those reported in similar behavioural intervention field experiments[18,27,97]. The participants were randomly assigned to one of two conditions, receiving either generic communication (control, $n = 6,524$) or communication that included a proximal cue (treatment, $n = 6,461$). To check the validity of the random assignment, we compared the distribution of demographic and financial characteristics between conditions and found that the groups were indeed balanced (see Table 1 for a summary of these characteristics).

### Procedure

The experiment included email communication that was sent to participating bank customers starting on 12 December 2023, ahead of the summer season in Australia. The objective of the communication was to urge customers to take steps to ensure their home was prepared for extreme weather-related events, particularly bushfires. Participants in both experimental conditions (treatment and control) were sent the same email, with the only difference being the subject line and the email's header. For the treatment group, those contained the specific name of the participant's suburb ('<first name>, prepare ahead to keep your property in <suburb> safe' and 'prepare your property in <suburb> in case of bushfires'). Whereas for the control group, the subject line and header were generic, not containing any information on the specific suburb ('<first name>, prepare ahead to help keep your property safe' and 'prepare your property in case of bushfires'). The content of the email included a short introduction, followed by three recommended steps for bushfire property preparation: clearing out gutters of leaves and debris that can quickly ignite, keeping a well-maintained lawn to minimize fire spread and removing inflammable items that can serve as potential fuel for bushfires. The email ended with a call-to-action button leading to a landing webpage on the bank's website containing additional information on property bushfire preparation. Importantly, the body of the email contained information on the consumer's suburb, for participants in both conditions, as part of the introduction ('according to the NSW Rural Fire Service, your property in <suburb> may be located in a bushfire prone area'). See Appendices A and B in Supplementary Information for full intervention texts and images, respectively, and Appendix C for the landing page image.

### Reporting summary

Further information on research design is available in the Nature Portfolio Reporting Summary linked to this article.

## Data availability

The data analysed in this article were provided by an Australian Bank and is proprietary. Under the terms of the bank's data-use agreement and applicable privacy laws, we are not authorized to publicly share the data. Michael Hiscox (hiscox@fas.harvard.edu), who manages the contractual relationship between Harvard University and the bank, can be contacted to facilitate requests to the bank for access to the archived data which will be contingent upon approval of the bank's research team. Alternatively, a replication package with a coarsened synthetic dataset would be provided in lieu of the original data.

## Code availability

All analyses reported in this study used the statistical software R (version 4.3.1). All R files are available via ResearchBox at https://research-box.org/2804.

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

## Acknowledgements

The authors are grateful to the teams at our field partner for their collaboration and partnership in designing and executing this field experiment. The authors received no specific funding for conducting this work. Open-access publication of this article was generously financed by the Stockholm School of Economics.

## Author contributions

N.N. conceptualized and designed the field experiment, coordinated its implementation, conducted the data analysis and wrote and revised the manuscript. M.H. secured and facilitated the partnership with the field partner, provided feedback on the study design and contributed to supervision.

## Competing interests

The authors declare no competing interests. The retail bank providing the data was given an opportunity to review manuscript content for factual inaccuracies but had no role in study design, data analysis, decision to publish or preparation of the manuscript.

## Additional information

**Correspondence and requests for materials** should be addressed to Nurit Nobel.

# Reporting Summary

## Statistics

For all statistical analyses, confirm that the following items are present in the figure legend, table legend, main text, or Methods section.

| n/a | Confirmed | |
|---|---|---|
| ☐ | ☒ | The exact sample size (*n*) for each experimental group/condition, given as a discrete number and unit of measurement |
| ☐ | ☒ | A statement on whether measurements were taken from distinct samples or whether the same sample was measured repeatedly |
| ☐ | ☒ | The statistical test(s) used AND whether they are one- or two-sided *Only common tests should be described solely by name; describe more complex techniques in the Methods section.* |
| ☒ | ☐ | A description of all covariates tested |
| ☒ | ☐ | A description of any assumptions or corrections, such as tests of normality and adjustment for multiple comparisons |
| ☐ | ☒ | A full description of the statistical parameters including central tendency (e.g. means) or other basic estimates (e.g. regression coefficient) AND variation (e.g. standard deviation) or associated estimates of uncertainty (e.g. confidence intervals) |
| ☐ | ☒ | For null hypothesis testing, the test statistic (e.g. *F*, *t*, *r*) with confidence intervals, effect sizes, degrees of freedom and *P* value noted *Give P values as exact values whenever suitable.* |
| ☒ | ☐ | For Bayesian analysis, information on the choice of priors and Markov chain Monte Carlo settings |
| ☒ | ☐ | For hierarchical and complex designs, identification of the appropriate level for tests and full reporting of outcomes |
| ☐ | ☒ | Estimates of effect sizes (e.g. Cohen's *d*, Pearson's *r*), indicating how they were calculated |

*Our web collection on statistics for biologists contains articles on many of the points above.*

## Software and code

Policy information about availability of computer code

| Data collection | Experimental data was received from a large Australian Bank under a Research Partnership Agreement with Harvard STAR Lab. |
|---|---|
| Data analysis | Data analysis was conducted in the statistical software R (v.4.3.1). Code is available publicly at Research Box: https://researchbox.org/2804 |

For manuscripts utilizing custom algorithms or software that are central to the research but not yet described in published literature, software must be made available to editors and reviewers. We strongly encourage code deposition in a community repository (e.g. GitHub). See the Nature Portfolio guidelines for submitting code & software for further information.

## Data

Policy information about availability of data

All manuscripts must include a data availability statement. This statement should provide the following information, where applicable:
- Accession codes, unique identifiers, or web links for publicly available datasets
- A description of any restrictions on data availability
- For clinical datasets or third party data, please ensure that the statement adheres to our policy

The data analyzed in this article were provided by the research partner (a large retail bank) and contain sensitive financial information. To protect participant privacy, and under a nondisclosure agreement, we are not authorized to publicly share the data. Under the terms of the bank's data-use agreement and applicable privacy laws, the underlying microdata cannot be shared with external parties. Interested researchers may contact the corresponding author at nurit.nobel@hhs.se

to obtain access to a replication package, with responses provided within 3 months. The package will include a synthetic dataset with coarsened data that mimics the structure and key summary statistics of the original data while containing no real customer records.

# Research involving human participants, their data, or biological material

Policy information about studies with [human participants or human data](). See also policy information about [sex, gender (identity/presentation), and sexual orientation]() and [race, ethnicity and racism]().

| | |
|---|---|
| Reporting on sex and gender | Descriptive statistics are reported for gender as reported by participants upon enrolling as bank customers. |
| Reporting on race, ethnicity, or other socially relevant groupings | Race or ethnicity data was not collected in this experiment and hence not reported in the manuscript. |
| Population characteristics | See "Setting and Participants" and Table 1 for detailed participant characteristics. |
| Recruitment | See "Setting and Participants" for inclusion criteria in the sample. |
| Ethics oversight | The Harvard University Committee on the Use of Human Subjects (CUHS) approved the protocol of this study (reference #: IRB23-1716) |

Note that full information on the approval of the study protocol must also be provided in the manuscript.

# Field-specific reporting

Please select the one below that is the best fit for your research. If you are not sure, read the appropriate sections before making your selection.

☐ Life sciences   ☒ Behavioural & social sciences   ☐ Ecological, evolutionary & environmental sciences

For a reference copy of the document with all sections, see [nature.com/documents/nr-reporting-summary-flat.pdf](http://nature.com/documents/nr-reporting-summary-flat.pdf)

# Behavioural & social sciences study design

All studies must disclose on these points even when the disclosure is negative.

| | |
|---|---|
| Study description | This is a field experiment, also known as Randomized Controlled Trial. It is a quantitative study. |
| Research sample | Participants were bank customers of our field partner, who have an active bank account (tenure with the bank between 1-64 years, M = 21.4, SD = 11.8), and who are homeowners whose property is located in an area determined to be prone to bushfires (wildfires) in New South Wales (NSW), Australia. Participants were between 19-96 years of age (M = 47.1, SD = 12.5), and 49.4% were female. |
| Sampling strategy | Bushfire risk was determined by the Rural Fire Service NSW Bushfire Prone Lands dataset, which is publicly available (State Government of NSW and NSW Rural Fire Service, 2023). |
| Data collection | The data analyzed in this article includes behavioral information (clicks, webpage visits) that were collected automatically and were provided by the research partner (a large bank). The experiment was blinded as participants were unaware that multiple message versions existed and thus unaware of their own treatment assignment. Randomization, intervention delivery, and outcome logging were automated by the partner platform and therefore blind. Data analysis was not performed blind to the conditions of the experiments. |
| Timing | Data was collected between December 14, 2023 and January 9, 2024 |
| Data exclusions | There were no post-data collection exclusions. |
| Non-participation | The bank had applied various exclusion criteria prior to sample selection (e.g. excluding customers who are minors, deceased, had previously opted out of marketing communications, participating in other bank campaigns, in hardship / arrears, guarantors). These customers were not part of the experiment, as pre-registered. |
| Randomization | Participants were randomly assigned to one of two conditions, receiving either generic communication (control, n = 6,524) or communication that included a proximal cue (treatment, n = 6,461). To check the validity of the random assignment, we compared the distribution of demographic and financial characteristics between conditions and found that the groups were indeed balanced (see Table 1 for a summary of these characteristics). |

# Reporting for specific materials, systems and methods

We require information from authors about some types of materials, experimental systems and methods used in many studies. Here, indicate whether each material, system or method listed is relevant to your study. If you are not sure if a list item applies to your research, read the appropriate section before selecting a response.

## Materials & experimental systems

| n/a | Involved in the study |
|-----|----------------------|
| ☒ ☐ | Antibodies |
| ☒ ☐ | Eukaryotic cell lines |
| ☒ ☐ | Palaeontology and archaeology |
| ☒ ☐ | Animals and other organisms |
| ☒ ☐ | Clinical data |
| ☒ ☐ | Dual use research of concern |
| ☒ ☐ | Plants |

## Methods

| n/a | Involved in the study |
|-----|----------------------|
| ☒ ☐ | ChIP-seq |
| ☒ ☐ | Flow cytometry |
| ☒ ☐ | MRI-based neuroimaging |

## Plants

| Seed stocks | NA |
|-------------|-----|
| Novel plant genotypes | NA |
| Authentication | NA |

