## [Peer Review File · Nature Human Behaviour]

Enhancing Climate Resilience with Proximal Cues in Personalized Climate Disaster Preparedness Messaging

Corresponding Author: Dr Nurit Nobel

Version 0:

Decision Letter:

24th March 2025

Dear Dr Nobel,

Thank you once again for your manuscript, entitled "Close to Home: Using Proximal Cues to Enhance Climate Resilience", and for your patience during the peer review process.

Your Article has now been evaluated by 3 referees. You will see from their comments copied below that, although they find your work of potential interest, they have raised quite substantial concerns. In light of these comments, we cannot accept the manuscript for publication, but would be interested in considering a revised version if you are willing and able to fully address reviewer and editorial concerns.

We hope you will find the referees' comments useful as you decide how to proceed. If you wish to submit a substantially revised manuscript, please bear in mind that we will be reluctant to approach the referees again in the absence of major revisions. We are committed to providing a fair and constructive peer-review process. Do not hesitate to contact us if there are specific requests from the reviewers that you believe are technically impossible or unlikely to yield a meaningful outcome.

To guide the scope of the revisions, the editors discuss the referee reports in detail within the team, including with the chief editor, with a view to (1) identifying key priorities that should be addressed in revision and (2) overruling referee requests that are deemed beyond the scope of the current study. We hope that you will find the prioritised set of referee points to be useful when revising your study. Please do not hesitate to get in touch if you would like to discuss these issues further.

As you can see, Reviewer 1 is concerned about the lack of theoretical advance over the existing literature, as well as the strength of the results. Reviewer 3 points out the very small effect sizes, which raise questions about the practical significance of your findings. The reviewers also point out that your measures are weak proxies of actual preparedness. In revision, it is important to address these concerns thoroughly, clarifying the advance your work represents over the existing literature and the practical relevance of your findings. (Please note that we do not require a lower statistical significance threshold for studies with large sample sizes; however, we do expect that effect sizes be thoroughly discussed.)

If you wish to submit a suitably revised manuscript, we would hope to receive it within 3 months. I would be grateful if you could contact us as soon as possible if you foresee difficulties with meeting this target resubmission date.

- Include a "Response to the editors and reviewers" document detailing, point-by-point, how you addressed each editor and referee comment. If no action was taken to address a point, you must provide a compelling argument. When formatting this document, please respond to each reviewer comment individually, including the full text of the reviewer comment verbatim followed by your response to the individual point. This response will be used by the editors to evaluate your revision and sent back to the reviewers along with the revised manuscript.
- Highlight all changes made to your manuscript or provide us with a version that tracks changes.

Link Redacted

Thank you for the opportunity to review your work. Please do not hesitate to contact me if you have any questions or would like to discuss the required revisions further.

Sincerely,

[Redacted Signature]

Nature Human Behaviour

Reviewer expertise:

Reviewer #1: climate change adaptation, environmental psychology, natural experiment

Reviewer #2: climate change adaptation, psychological distance, field experiment

Reviewer #3: climate change, environmental psychology

REVIEWER COMMENTS:

Reviewer #1 (Remarks to the Author):

Thanks so much for the opportunity to review this piece! The authors addressed a crucial issue of how to enhance climate resilience among people who are vulnerable to climate change disasters like bushfires.

This manuscript is well written, the methodological procedures and details are all clear, and great to see the authors using a large online survey experiment (I would prefer it more than the field experiment) that increased tangibility (i.e., reduced psychological distance) of climate communications via using the local suburb information. I enjoyed reading the whole piece, so thank you so much.

This work bears very promising practical implications for climate risk communication and climate adaptation, however, is theoretically not new in contributions. As an environmental social psychologist, I am not surprised at all to see how reducing psychological distance and evoking place attachment can work to promote either climate change mitigation or adaptation - those intervention approaches increase the salience of self-relevance so that the risk of climate change inactions is directly relevant to the self, which help promote behaviour uptake. That said, to claim the first work to close the gap by using a behavioural intervention to encourage climate resilience is somewhat thin in theoretical contributions.

The great value I easily get is the use of a large experiment with 12,985 participants recruited by [Redacted]. Yet, with this large sample size, the P value would be sensitively leaning toward indicating significant differences, even though the practical effect size could be small or negligible. In this paper, the p values for landing page clicks, landing page visits, and email openings were = 0.001, 0.005, and 0.049, respectively. If a stricter p value threshold ($p < 0.001$) was applied with such a large sample size, none of these outcome variables would suggest sig differences between the control and experiment conditions.

Also, a bit down to read the Discussion section, as I expected there would be in-depth discussions on results but turned out that this part only summarised the results against the literature mentioned in the Introduction. And it would be better if this work also reflected on behavioural interventions used in the wild, whether it works, how to scale up, etc.

Reviewer #2 (Remarks to the Author):

This work addresses a timely issue and does so with an applied focus, recruiting a large-N sample in Australia, and using an online email experiment to influence climate resilience behavior for a low-salient future threat. I applaud the authors on a robust study, that is pre-registered, and a resulting manuscript that is, in my opinion, coherently written.

Introduction

I found the introduction to be well-written. It offers a good overview of the literature on field interventions in the realm of climate change, and skillfully intertwines this literature with theory from psychological distance and construal theory.

However, there are a few places and a few key references that I think are missing.

The first is this recent paper on how psychological distance is overestimated and not as predictive of climate outcomes. I have my issues with this paper (e.g., a majority of people still find climate change/global warming distant, and there are multiple forms of psychological distance), but I think it requires acknowledgement and critical discussion in the Introduction.

van Valkengoed, A. M., Steg, L., & Perlaviciute, G. (2023). The psychological distance of climate change is overestimated. *One Earth*, 6(4) 362-391. <https://doi.org/10.1016/j.oneear.2023.03.006>

The second is a brief reference and acknowledgement of strategies that have proven effective at reducing psychological distance. In my opinion, this distance stems from both the social and the temporal component of climate outcomes (we think they will affect others in the future). In the Vlasceanu et al. paper the authors discuss (but also in subsequent papers from that groups which re-analyze the data including national and person level moderators) interventions focusing on intergenerational values, or future-self continuity, which seek to also reduce psychological distance proved very effective as well. I think it's worth highlighting this as it further supports the authors' argument. There is additional literature on this subject, but I think adding that might detract from the flow of the Introduction, Thus, I think explaining this in greater detail using existing literatures and work from Vlasceanu and colleagues helps, and does not burden the Intro further.

Results

Results were easy to follow and I want to applaud the authors on a their work ensuring that they are presented coherently and in a very digestible manner. My only comments here concern the figures. I think the authors should use 95% C.I.s rather than standard errors, and for figure 3, the asterisk detailing significance should be removed, given the non-significant (or marginally-significant) result. Further, what could be helpful is the provision of information about the overall tendency to open emails/click through websites in the literature, to situate the very small number of people who opted to do this behavior. I assume the N is usually small, but it might be good to situate this in an existing pattern of results for similar outcomes.

Discussion

Very little to add here, Overall nice work. My only comment concerns whether the authors would like to speculate about what additional parameters (e.g., localized climate threat information) could have helped increase the effectiveness of the manipulation. How could future research further add and potentially increase the effectiveness of this low-cost intervention?

Methods

No comment.

Other Comments

I am curious about moderation effects. Was the message more or less effective based on age, gender, income or financial information? It would be good to highlight this and place it in the supplementary materials.

Reviewer #3 (Remarks to the Author):

This is a well-written and executed large scale field experiment involving over 12,000 homeowners in wildfire-prone areas of Australia. The study found that adding a proximal cue (i.e., the homeowner's suburb name) to emails increased engagement with wildfire preparedness information. Relative to the control group, the treatment group scored higher on all engagement metrics (clicks, visits, opens).

The methodology is strong: large sample, randomised experimental design, and multiple behavioural DVs (as opposed to self-report). The study also has a strong theoretical grounding in Construal Level Theory and the psychological distance literature.

Nevertheless there are two notable shortcomings, both acknowledged by the authors, relevant to the practical relevance of these findings.

First, even though the click-through-rate doubled in the treatment condition relative to the control, the click and visit rates were less than 1% for participants in the treatment condition. So whereas the impact of the treatment seems impressive in odds ratio terms, it's quite tiny in practical terms, at least in terms of impacting individual behaviour.

Second, the DVs used in the study (clicks and visits) are proxy measures for actual risk mitigation behaviour, which is not directly assessed. This leaves a pretty big gap between message engagement (clicks) and tangible preparedness.

On the plus side the intervention is simple and inexpensive to implement, and That limits the practical impact per individual, though in aggregate it might be meaningful.

Overall, I think the study is worth publishing. It provides a good example of a well-designed large scale field experiment assessing the impact of a simple messaging treatment on behavioural (albeit proxy) outcomes, while also illustrating that cost effective, scalable interventions may be worth implementing even if the effects on individual behaviour is quite small.

Version 1:

Decision Letter:

Our ref: NATHUMBEHAV-24125238A

3rd July 2025

Dear Dr. Nobel,

Thank you for submitting your revised manuscript "Enhancing Climate Resilience with Proximal Cues in Personalized Climate Disaster Messaging" (NATHUMBEHAV-24125238A). It has now been seen by the original referees and their comments are below. As you can see, the reviewers find that the paper has improved in revision. We will therefore be happy in principle to publish it in Nature Human Behaviour, pending minor revisions to satisfy the referees' final requests and to comply with our editorial and formatting guidelines.

We are now performing detailed checks on your paper and will send you a checklist detailing our editorial and formatting requirements within two weeks. Please do not upload the final materials and make any revisions until you receive this additional information from us.

Sincerely,

Nature Human Behaviour

Reviewer #1 (Remarks to the Author):

This revised version is considerate and thorough in terms of the specific theoretical contributions to the debate of psychological distance intervention and within the climate adaptation field.

I also like that the authors reflected and discussed in depth about this contextualised and personalised intervention against the broad literature of behavioural interventions in the literature and in the wild.

With this said, I have no further concerns or issues with this revised version. Best wishes!

Reviewer #2 (Remarks to the Author):

I believe that the authors have addressed my comments with a lot of care and rigor.

Reviewer #3 (Remarks to the Author):

I commend the authors for conducting an excellent study and constructively responding to reviewer feedback.

Response to reviewers – Manuscript ID NATHUMBEHAV-24125238

We greatly appreciate the thoughtful feedback provided by the reviewers and thank the editors for giving us the opportunity to submit this revision. Here is an overview of key changes we made in the revised manuscript:

- **Clarified the theoretical contribution** by situating our work within recent debates on psychological distance and behavioral interventions.
- **Addressed concerns about practical significance** by contextualizing small effect sizes and expanding the discussion on their implications for real-world impact, and by discussing engagement as a necessary pre-cursor to action.
- **Improved transparency and robustness** by including additional analyses in the Supplementary Information, including moderation tests, and updating all figures to display 95% confidence intervals and remove asterisks.
- **Reformatted the manuscript to align with the revision checklist**, including changing title, reordering sections, updating figure notations, and ensuring consistency with Nature Human Behaviour’s style and editorial policies

We are grateful for the insightful suggestions made by the reviewers and are hopeful that the reviewers and editors would feel that we made them justice with this revision. Please see below detailed answers for each reviewer’s comments. Thank you.

Responses to reviewer 1

	Comment from reviewer	Response	Changes / additions to manuscript
1-1	This work ... is theoretically not new in contributions. As an environmental social psychologist, I am not surprised at all to see how reducing psychological distance and evoking place attachment can work to promote either climate change mitigation or adaptation.... That said, to claim the first work to close the gap by using a behavioural intervention to encourage climate resilience is somewhat thin in theoretical contributions.	We thank the reviewer for this important feedback. In the revised manuscript, we aimed to clarify and strengthen our theoretical contribution as follows. While some in the field may find the effects of reducing psychological distance intuitive, recent literature has questioned the robustness of such effects. For example, van Valkengoed et al. (2023) argue that psychological distance is often overestimated and may not consistently predict climate-related outcomes, and Brügger et al. (2015) argue that psychological distance-based interventions are not always effective. In the revised manuscript, we explicitly address this debate, with our study supporting the view that	1) Added paragraph in the Introduction section on the debate around the efficacy of psychological distance (see also 2-1). The paragraph starts with “The idea that psychological distance is a central barrier ...” and is highlighted. 2) Clarified the domain novelty in the relevant paragraph in the Introduction section. Paragraph starts with “Despite the promise

		psychological distance-based interventions can be efficacious to drive climate action. Additionally, we believe that our manuscript explores the effect of psychological distance in a novel context. Existing behavioral science research on psychological distance has overwhelmingly focused on climate mitigation (e.g., carbon reduction, sustainable consumption). To our knowledge, no prior field experiments have applied this framework to the domain of climate adaptation—particularly to protective homeowner behaviors in the face of acute environmental risks. Our study aims to fill this gap by demonstrating that these same mechanisms can be successfully extended to climate resilience behavior. We’ve made adjustments in the text of the introduction to clarify this. Having said this, we accept the critique regarding overclaiming novelty, and in line with journal guidelines we toned down such claims throughout the manuscript. References Brügger, A., Dessai, S., Devine-Wright, P., Morton, T. A., & Pidgeon, N. F. (2015). Psychological responses to the proximity of climate change. Nature climate change, 5(12), 1031-1037. Van Valkengoed, A. M., Steg, L., & Perlaviciute, G. (2023). The psychological distance of climate change is overestimated. One Earth, 6(4), 362-391.	that using proximal cues ...” and changes are highlighted. 3) Modified theoretical contribution paragraph in the Discussion section. The paragraph starts with “This study makes three key contributions.” and is highlighted. 4) Softened novelty and gap-closing claims throughout the manuscript (highlighted)
1-2	With this large sample size, the P value would be sensitively leaning toward indicating significant differences, even though the practical effect size could be small or negligible. In this paper, the p values for landing page clicks, landing page	In light of the editor’s note regarding statistical significance, we will focus this response on addressing the issue of small effect size. We thank the reviewer for this valid point, which was also raised by Reviewer #3 (see 3-1). The reviewer is correct that while the odds of clicking were about twice as high in the treatment group (OR = 2.03), the absolute	Added a paragraph discussing the meaning of the effect sizes in the Discussion section. The paragraph starts with “Although the increase in engagement ...” and is highlighted.

visits, and email openings were = 0.001, 0.005, and 0.049, respectively. If a stricter p value threshold ($p < 0.001$) was applied with such a large sample size, none of these outcome variables would suggest significant differences between the control and experiment conditions. Related note from editor: Please note that we do not require a lower statistical significance threshold for studies with large sample sizes; however, we do expect that effect sizes be thoroughly discussed.	difference in click rates was modest (0.959% vs. 0.475%). While OR is the standard effect size measure in logistic models and binary outcomes, it is true that it can appear large even when the underlying probabilities are small. That is the case here—our effect is statistically robust but small in absolute terms. Still, we believe the effect is meaningful. As argued by Götz et al. (2021) and Matz et al. (2017) (among others), focusing exclusively on large effect size hinders a nuanced exploration of complex psychological phenomena, and risks overlooking effects that are likely to be real. In the case of our study, it involved a field experiment with nearly 13,000 real individuals, and such designs typically yield smaller but more ecologically valid effects than controlled lab (or online) studies. Lastly, even small effect sizes can have a large impact when evaluated over time or at scale. In behavioral interventions, even minor nudges can accumulate—across time, across populations, and across policy channels. When scaled, a small change in behavior (like clicking a link or visiting a site) can translate into substantial collective action. This is especially true in the context of climate adaptation, where mobilizing even a slightly larger share of the public can make a measurable difference. To illustrate, the observed difference of 0.484 percentage points in our study translates into an additional 25,000 individuals accessing the wildfire mitigation information if the message were scaled to 5 million bank customers ($0.00484 \times 5,000,000$). If similar messaging was adopted by other banks or public agencies and distributed nationally to approximately 20 million Australian adults (ABS, 2024), this would amount to an additional 96,800 people engaging	
---	---	--

		with wildfire mitigation recommendations. These figures highlight that even marginal improvements in engagement can yield substantial real-world impact when interventions are deployed at population scale. We welcome the opportunity to highlight practical relevance of the effect size in the paper, and have added this in the Discussion section. To avoid the risk of overclaiming, we have not included the extrapolation example above in the manuscript. We would be happy to incorporate it if the reviewer or editors feel it would add value. References: Australian Bureau of Statistics. (2024). National, state and territory population. https://www.abs.gov.au/statistics/people/population/national-state-and-territory-population/latest-release Götz, F. M., Gosling, S. D., & Rentfrow, P. J. (2022). Small effects: The indispensable foundation for a cumulative psychological science. Perspectives on psychological science, 17(1), 205-215. Matz, S. C., Gladstone, J. J., & Stillwell, D. (2017). In a world of big data, small effects can still matter: A reply to Boyce, Daly, Hounkpatin, and Wood (2017). Psychological science, 28(4), 547-550.	
1-3	[In] the Discussion section, as I expected there would be in-depth discussions on results but turned out that this part only summarised the results against the literature mentioned in the Introduction. And it would be better if this work also reflected on behavioural interventions	We appreciate the suggestion to expand the discussion of behavioral interventions. We agree that this is an important area deserving deeper consideration. In response, we have substantially revised the relevant paragraph in the Discussion section to reflect more fully on the application of behavioral interventions in real-world contexts.	Revised the paragraph on behavioral interventions in the Discussion section. The paragraph starts with “Second, this study advances the understanding of the applicability of behavioral interventions ...” and is highlighted.

	used in the wild, whether it works, how to scale up, etc.		
--	---	--	--

Responses to reviewer 2

	Comment from reviewer	Response	Changes / additions to manuscript
2-1	There are a few places [in the introduction] and a few key references that I think are missing. The first is this recent paper on how psychological distance is overestimated and not as predictive of climate outcomes. I have my issues with this paper (e.g., a majority of people still find climate change/global warming distant, and there are multiple forms of psychological distance), but I think it requires acknowledgement and critical discussion in the Introduction. van Valkengoed, A. M., Steg, L., & Perlaviciute, G. (2023). The psychological distance of climate change is overestimated. One Earth, 6(4) 362-391.	We thank the reviewer for highlighting the van Valkengoed et al. paper. We agree that it adds nuance to the discussion on the potential efficacy of psychological distance interventions for encouraging climate action, and merits a more thorough discussion. We have incorporated it into the introduction.	Added a paragraph to discuss the critique of psychological distance interventions in the Introduction section. The added paragraph starts with the sentence “The idea that psychological distance is a central barrier to climate action has been recently called into question.” And has been highlighted in the revised manuscript.
2-2	The second is a brief reference and acknowledgement of strategies that have proven effective at reducing psychological distance. In my opinion, this distance stems from	We agree that a deeper discussion of the Vlasceanu et al. paper (and subsequent work) provides further support for our argument in favor of psychological distance interventions. We incorporated this discussion into the same paragraph addressing 2-1 that was added to the introduction.	See 2-1.

	both the social and the temporal component of climate outcomes (we think they will affect others in the future). In the Vlasceanu et al. paper the authors discuss (but also in subsequent papers from that groups which re-analyze the data including national and person level moderators) interventions focusing on intergenerational values, or future-self continuity, which seek to also reduce psychological distance proved very effective as well. I think it's worth highlighting this as it further supports the authors' argument. There is additional literature on this subject, but I think adding that might detract from the flow of the Introduction, Thus, I think explaining this in greater detail using existing literatures and work from Vlasceanu and colleagues helps, and does not burden the Intro further.		
2-3	Results were easy to follow and I want to applaud the authors on a their work ensuring that they are presented coherently and in a very digestible manner. My only comments here concern the figures. I think the authors should use 95% C.I.s rather than standard errors.	We thank the reviewer for the positive feedback. As for the error bars, we agree that confidence intervals provide more informative representations of statistical uncertainty, and have therefore replaced error bars in all figures with 95% confidence intervals instead of standard errors.	Replaced Figures 1-3 in the Results Section.

2-4	and for figure 3, the asterisk detailing significance should be removed, given the non-significant (or marginally-significant) result.	We thank the reviewer for pointing this out. In line with the reviewer’s suggestion, and consistent with Nature Human Behaviour’s emphasis on transparent reporting, we have removed all asterisks from Figures 1-3. Given that exact p-values and effect sizes are already reported throughout the manuscript and supplementary materials, we felt this graphic representation was superfluous. This change aligns with current best practices in statistical reporting.	Replaced Figures 1-3 in the Results Section.
2-5	what could be helpful is the provision of information about the overall tendency to open emails/click through websites in the literature, to situate the very small number of people who opted to do this behavior. I assume the N is usually small, but it might be good to situate this in an existing pattern of results for similar outcomes.	The reviewer suggests to contextualize our click-through numbers using external benchmarks. We agree that this could provide useful perspective. However, after reviewing available sources, we found that most click-through data originates from industry reports produced by email marketing platforms, and these vary widely in methodology, metrics, and scope. Moreover, benchmarks differ substantially across sectors and campaign types. In our case, while the message was distributed by a bank (therefore classified as pertaining to the “Financial Services” industry), it was not a typical banking email, but rather an informational message about climate risk. This makes it particularly difficult to identify a suitable benchmark. Industry sources generally suggest that informational emails—especially those sent at scale to broad audiences—tend to yield lower engagement than promotional emails targeted to specific customer needs. So, while we believe that our findings are broadly consistent with these general trends, we hesitate to include specific CTR comparisons in the manuscript given the lack of credible, robust, and relevant benchmarks. Instead, we point to our methodological contextualization of the effect size, see 1-2.	See 1-2.

2-6	[Discussion] Very little to add here, Overall nice work. My only comment concerns whether the authors would like to speculate about what additional parameters (e.g., localized climate threat information) could have helped increase the effectiveness of the manipulation. How could future research further add and potentially increase the effectiveness of this low-cost intervention?	Thank you for this excellent suggestion. We agree that incorporating additional localized risk information could enhance the intervention and merits studying. We have added this point to the future research section in the Discussion.	Text added in the paragraph discussing limitations and future research directions in the Discussion section. The relevant part starts with “Additionally, future research could explore the integration of more specific, localized climate risk information...” and is highlighted.
2-7	I am curious about moderation effects. Was the message more or less effective based on age, gender, income or financial information? It would be good to highlight this and place it in the supplementary materials.	We conducted a series of moderation analyses examining whether the effect of the message varied by age, gender, income, and balance across various accounts. None of the interaction effects were statistically significant, suggesting that the message was similarly effective across subgroups. We report these results in the Supplementary Information (Appendix E).	Added Appendix E: Exploring moderation effects – regression results in the Supplementary Information.

Responses to reviewer 3

	Comment from reviewer	Response	Changes / additions to manuscript
3-1	The methodology is strong ... Nevertheless there are two notable shortcomings, both acknowledged by the authors, relevant to the practical relevance of these findings. First, even though the click-through-rate doubled in the treatment condition relative to the control, the	See our response to this valid critique in 2-1.	See 2-1.

	click and visit rates were less than 1% for participants in the treatment condition. So whereas the impact of the treatment seems impressive in odds ratio terms, it's quite tiny in practical terms, at least in terms of impacting individual behaviour.		
3-2	Second, the DVs used in the study (clicks and visits) are proxy measures for actual risk mitigation behaviour, which is not directly assessed. This leaves a pretty big gap between message engagement (clicks) and tangible preparedness.	The reviewer raises an important point. We recognize that engagement is not equivalent to behavior, and we acknowledge this as a limitation of our study in the discussion. At the same time, we view engagement as a meaningful precursor to action. To clarify this, we have added a brief theoretical rationale in the relevant paragraph in the Discussion section.	A sentence starting with “With that said, engagement—such as clicking on an informational link—is widely recognized...” has been added to the limitations paragraph in the Discussion section and is highlighted.
3-3	Overall, I think the study is worth publishing. It provides a good example of a well-designed large scale field experiment assessing the impact of a simple messaging treatment on behavioural (albeit proxy) outcomes, while also illustrating that cost effective, scalable interventions may be worth implementing even if the effects on individual behaviour is quite small.	Thank you for your supportive feedback.	NA